# Operationalising the Superficial Alignment Hypothesis via Task Complexity

**Tomás Vergara-Browne** [1 2]   **Darshan Patil** [1 3]   **Ivan Titov** [4 5]   **Siva Reddy** [1 2 6]
**Tiago Pimentel** [* 7]   **Marius Mosbach** [* 1 2]

## Abstract

The superficial alignment hypothesis (SAH) posits that large language models learn most of their knowledge during pre-training, and that post-training merely surfaces this knowledge.[1] The SAH, however, lacks a precise definition, which has led to (i) different and seemingly orthogonal arguments supporting it, and (ii) important critiques to it. We propose a new metric called **task complexity**: the length of the shortest program that achieves a target performance on a task. In this framework, the SAH simply claims that pre-trained models drastically reduce the complexity of achieving high performance on many tasks. Our definition unifies prior arguments supporting the SAH, interpreting them as different strategies to find such short programs. Experimentally, we estimate the task complexity of mathematical reasoning, machine translation, and instruction following; we then show that these complexities can be remarkably low when conditioned on a pre-trained model. Further, we find that pre-training enables access to strong performances on our tasks, but it can require programs of gigabytes of length to access them. Post-training, on the other hand, collapses the complexity of reaching this same performance by several orders of magnitude. Overall, our results highlight that task adaptation often requires surprisingly little information—often just a few kilobytes.

* Equal advising.   [1]Mila Quebec AI Institute [2]McGill University [3]Université de Montréal [4]University of Edinburgh [5]University of Amsterdam [6]Canada CIFAR AI Chair [7]ETH Zürich.   Correspondence to: Tomás Vergara-Browne <tomas.vergarabrowne@mila.quebec>, Tiago Pimentel <tiago.pimentel@inf.ethz.ch>, Marius Mosbach <marius.mosbach@mila.quebec>.

*Proceedings of the $43^{rd}$ International Conference on Machine Learning*, Seoul, South Korea. PMLR 306, 2026. Copyright 2026 by the author(s).

[1]To avoid confusion with the AI safety literature, we will say models are 'adapted', rather than 'aligned', to a task. We use the name 'superficial alignment hypothesis', however, as opposed to 'superficial adaptation hypothesis', due to historical reasons.

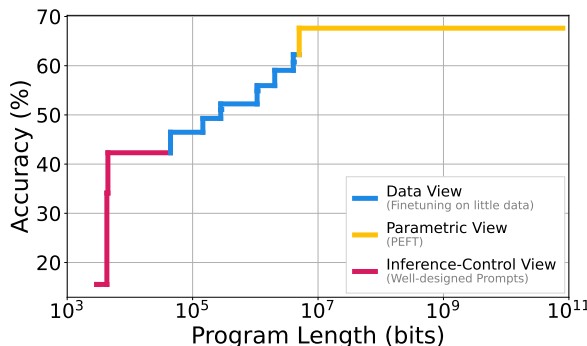

*Figure 1.* Pareto curve of program length vs. performance for Olmo3-7B on GSM8K. We argue that prior works about the superficial alignment hypothesis can be seen as proposing different approaches to find short programs to solve a task, and we find that these different views inform different regions of this Pareto curve.

## 1. Introduction

The superficial alignment hypothesis (SAH) states that a large language model's *"knowledge and capabilities are learned almost entirely during pre-training, while alignment teaches it which subdistribution of formats should be used when interacting with users"* (Zhou et al., 2023a). This hypothesis has been supported from different angles, e.g., we can adapt pre-trained models to solve complex tasks by: ① finetuning it on few examples (Zhou et al., 2023a; Ye et al., 2025), ② updating only a few of its parameters (Han et al., 2024; Li & Kim, 2024), or ③ using well-designed prompts (Lin et al., 2024; Lake et al., 2025).

Despite its appeal, however, the SAH suffers from imprecise definitions of important terms such as 'knowledge', 'capabilities', and 'subdistribution of formats'. This ambiguity has created two main problems. First, arguments supporting the hypothesis are seemingly orthogonal: some emphasise the small data requirements to adapt a model (① above), others the number of parameters needed to be updated (②), and still others the fact that adaptation can be achieved through inference-time control alone (③). This makes it unclear whether these reflect a common underlying phenomenon. Second, critics have exploited these undefined terms to challenge the hypothesis. Raghavendra et al. (2024) and Lambert (2025), for instance, interpret that if a model contains 'knowledge' and 'capabilities' for a task, its performance on the task should saturate very easily; by

showing that, on many tasks, performance saturation requires a significant amount of fine-tuning or post-training, they conclude the SAH is incomplete. These tensions highlight the need for a more precise operationalisation of the SAH that can bring clarity to this debate.

In this paper, we provide a clearer definition of the superficial alignment hypothesis, grounded in algorithmic information theory. We first define the **complexity of a task at performance** $\delta$ as the length of the shortest program to achieve performance $\delta$ on it. We then argue that the SAH can be operationalised as claiming that many complex tasks—i.e., tasks for which no short program exists which achieves a certain target performance on it—become low-complexity once we give our programs access to a pre-trained model. This perspective places seemingly orthogonal arguments supporting the SAH on a common, quantitative footing: arguments based on data efficiency, parameter efficiency, and inference control define different classes of programs whose lengths can be directly compared (see Figure 1). These arguments are, in fact, instances of the same underlying principle: there exist short programs to achieve target performance in the task. The arguments do not compete with each other, but rather simply use different strategies to find such programs.

Experimentally, we measure the task complexity of three diverse NLP tasks (mathematical reasoning, machine translation and instruction following) when given access to the weights of either: SmolLM3 3B (Bakouch et al., 2025), Olmo3 7B, or Olmo3 32B (Olmo et al., 2025). We find that programs as small as $1.2 \times 10^6$ bits (which is 151 kilobytes, or the size of a single image from ImageNet; Russakovsky et al., 2015) can adapt these large language models (LLMs) to achieve strong performance on some of these tasks.

We further analyse how task complexity evolves during training by examining randomly initialised, pre-trained, and post-trained checkpoints of SmolLM3 and Olmo3 7B. Strong performance on these tasks first becomes accessible with pre-trained checkpoints, but requires the use of long programs to plateau the obtained performance (typically on the order of megabytes to gigabytes). Post-trained checkpoints enable access to the same strong performance, however, with dramatically shorter programs.

Overall, our work provides a rigorous framework to understand the superficiality of LLM adaptation from an algorithmic information perspective: adapting a pre-trained model for a task often requires just kilobytes of information.

## 2. Perspectives on Superficial Adaptation

We present 3 different views from prior work which highlight the idea that knowledge about a task is already present in pre-trained models, and not learned at later adaptation stages. These are: the data view, the parametric view, and the inference-control view.

① **Data view.** *Adaptation is superficial because we can finetune an LLM on few data points to adapt it to a new task.*

Adapting a pre-trained LLM to a task requires remarkably little data in comparison to the amount of data used in pre-training (Zhou et al., 2023a; Qi et al., 2024; Ye et al., 2025; Muennighoff et al., 2025). For instance, Zhou et al. (2023a) show that 1000 examples are enough to adapt a pre-trained LLM into a strong instruction-following model. When adopting the data view, the contrast between the small amount of data necessary for adaptation and the much larger amounts of data used in pre-training can be interpreted as evidence for the superficiality of adaptation.

② **Parametric view.** *Adaptation is superficial because only few parameters need to be changed to adapt an LLM to a new task.*

There is ample evidence showing that fine-tuning a pre-trained LLM requires a surprisingly small amount of trainable parameters (Li & Liang, 2021; Hu et al., 2022; Liu et al., 2022a; Ben Zaken et al., 2022; Liu et al., 2022b; Dettmers et al., 2023; Yadav et al., 2025; Han et al., 2024; Li & Kim, 2024; Ping et al., 2024; Liu et al., 2024; Huang et al., 2025; Morris et al., 2026). For example, LoRA (Hu et al., 2022) allows us to efficiently finetune LLMs by adding trainable adapters which represent a small fraction (less than 1%) of a model's parameters. The contrast between the small amount of trainable parameters during adaptation and the large amount of parameters needed in pre-training can be seen as evidence for the superficiality of adaptation.

③ **Inference-control view.** *Adaptation is superficial because we can keep all LLM weights frozen, and only make simple changes to its inference code to adapt it to a task.*

In some cases, it is not even necessary to modify the parameters of a pre-trained LLM at all to perform well on a downstream task (Brown et al., 2020; Lin et al., 2024; Hewitt et al., 2024; Minder et al., 2025; Braun et al., 2025; Chen et al., 2025; Karan & Du, 2025). For example, Lin et al. (2024) show how to carefully design a prompt which makes a pre-trained LLM follow instructions. Under this view, the fact that no weights have to be changed at all can be seen as evidence for the superficiality of adaptation.

## 3. The Superficial Alignment Hypothesis

While the previous three views may seem orthogonal, their results can still be interpreted as evidence of superficiality. Why is that? We argue that each of them evokes a sense that most of the information required to perform well on a task is already present in the pre-trained model. We ground this intuition from the perspective of algorithmic information theory (Li & Vitányi, 2008). These definitions will allow us to interpret each of the ① data view, ② parametric view, and ③ inference-control view, as short *programs* which

do not add a lot of information to a pre-trained model. We start by defining the concept of a task.

**Definition 3.1.** *Let a **task** $\mathtt{T}$ be a tuple $(\mathcal{X}, \mathcal{Y}, p, \mathcal{S})$, where:*

- $\mathcal{X}$ *and* $\mathcal{Y}$ *are, respectively, an input and output space.*

- $p : \mathcal{X} \to [0, 1]$ *is a probability distribution over inputs* $x \in \mathcal{X}$.

- $\mathcal{S} : \mathcal{X} \times \mathcal{Y} \to \mathbb{R}$ *is a scoring function which evaluates the quality of an input-output pair.*

Let us use the example of solving grade school math problems as our task $\mathtt{T}$. Both $\mathcal{X}$ and $\mathcal{Y}$ are the space of text strings, and $p$ is a distribution over grade school math problems written in text. For example, $p$ would assign a non zero probability to an $x \in \mathcal{X}$ such as *"how many fruits are three apples and two oranges?"*. The scoring function $\mathcal{S}$ takes a problem $x$ and an answer $y$, and returns 1 if $y$ correctly solves $x$, and 0 otherwise. So given a $y \in \mathcal{Y}$ of *"five"*, we would have $\mathcal{S}(x, y) = 1$.

Now, let $\mathrm{M}$ be a Universal Turing Machine. This machine takes in a program $\mathtt{P} \in \mathcal{P}$ and an input $x \in \mathcal{X}$, to compute an output $y \in \mathcal{Y}$.[2] We can write $\mathrm{M}(\mathtt{P}, x)$ as the output of running program $\mathtt{P}$ on input $x$ (if it halts). As throughout our discussion we keep $\mathrm{M}$ unspecified and fixed, we will use $\mathtt{P}(x)$ to represent $\mathrm{M}(\mathtt{P}, x)$. We can now define the score of a program $\mathtt{P}$ on a task $\mathtt{T}$ as its expected performance on it.

**Definition 3.2.** *Given a task $\mathtt{T} = (\mathcal{X}, \mathcal{Y}, p, \mathcal{S})$ and a program $\mathtt{P}$, we define the **score** of $\mathtt{P}$ on $\mathtt{T}$ as:*

$$\mathtt{score}_{\mathtt{T}}(\mathtt{P}) \stackrel{\text{def}}{=} \mathbb{E}_{x \sim p}[\mathcal{S}(x, \mathtt{P}(x))] \tag{1}$$

Coming back to our example, we can design a program $\mathtt{P}$ that achieves some score on task $\mathtt{T}$. The program $\mathtt{P}$ may include any logic necessary to output some response $y$ whenever an input $x$ is provided. The program must, however, be completely self-contained, meaning that it cannot access anything that is not directly encoded in $\mathtt{P}$ or $x$. The score of this program $\mathtt{P}$ is represented as $\mathtt{score}_{\mathtt{T}}(\mathtt{P})$, and we will say that program $\mathtt{P}$ *solves* $\mathtt{T}_{\delta}$ if $\mathtt{score}_{\mathtt{T}}(\mathtt{P}) \geq \delta$.

## 3.1. Task Complexity

We now characterise the complexity of a task $\mathtt{T}$ at performance $\delta$ as the length of the shortest program that solves it.

**Definition 3.3.** *The **complexity of task** $\mathtt{T}$ **at performance** $\delta$ is defined as:*

$$\mathrm{C}(\mathtt{T}_{\delta}) \stackrel{\text{def}}{=} \min_{\mathtt{P} \in \mathcal{P}} \{len(\mathtt{P}) : \mathtt{score}_{\mathtt{T}}(\mathtt{P}) \geq \delta\} \tag{2}$$

This metric quantifies the amount of information required to solve (or represent) task $\mathtt{T}$ at a certain performance level, being tightly related to both Kolmogorov complexity and rate-distortion theory (as further discussed in Section 3.3). If a $\mathtt{T}_{\delta}$ is less complex than another $\tilde{\mathtt{T}}_{\delta}$, then we can solve $\mathtt{T}_{\delta}$ with a shorter program than what we need to solve $\tilde{\mathtt{T}}_{\delta}$.

In our running example, $\mathrm{C}(\mathtt{T}_{90\%})$ would represent the length of the minimum program to solve grade school math problems with 90% accuracy. Presumably, this program might be very long, as it requires understanding natural language, planning the steps for which to solve the problem and then computing the answer. Given access to an LLM, however, even such a complex task might be greatly simplified. This motivates our definition of conditional task complexity: the length of the minimum program $\mathtt{P}$ which, given access to a model's weights $\theta$ (represented as a bit-string[3]), achieves a certain performance at task $\mathtt{T}$. We will use $\mathtt{P}_{\theta}$ to represent a program $\mathtt{P}$ with access to a model's weights $\theta$, where we treat this access to $\theta$ as an additional input of the program, i.e., $\mathtt{P}_{\theta}(x) = \mathtt{P}(x, \theta)$.

**Definition 3.4.** *The **complexity of task** $\mathtt{T}$ **at performance** $\delta$ **conditioned on** $\theta$ is defined as:*

$$\mathrm{C}(\mathtt{T}_{\delta} \mid \theta) \stackrel{\text{def}}{=} \min_{\mathtt{P} \in \mathcal{P}} \{len(\mathtt{P}) : \mathtt{score}_{\mathtt{T}}(\mathtt{P}_{\theta}) \geq \delta\} \tag{3}$$

While the task of solving grade school math problems with 90% accuracy is very complex, i.e., $\mathrm{C}(\mathtt{T}_{90\%})$ is presumably very high, access to a pre-trained model $\theta$ may lead to a considerably lower complexity $\mathrm{C}(\mathtt{T}_{90\%} \mid \theta)$. This reduction in complexity defines the amount of information that a model $\theta$ contains about a task $\mathtt{T}$ at performance $\delta$.

**Definition 3.5.** *The **information a model** $\theta$ **contains about a task** $\mathtt{T}$ **at performance** $\delta$ is defined as:*

$$\mathrm{I}(\mathtt{T}_{\delta}; \theta) \stackrel{\text{def}}{=} \mathrm{C}(\mathtt{T}_{\delta}) - \mathrm{C}(\mathtt{T}_{\delta} \mid \theta) \tag{4}$$

We derive a few theoretical properties of these proposed metrics in Appendix A, leveraging existing results in algorithmic information theory for related concepts.

## 3.2. Adaptability and SAH

So far, our definitions focused on tasks and their complexity. The SAH, however, is about pre-trained models, discussing how adaptable they are to new tasks. We thus now put forward a definition of how adaptable a model is to a task. Specifically, we say that a model is $(b, \delta)$-adaptable for a task $\mathtt{T}$ if there exists a program shorter than $b$ which allows the model to solve $\mathtt{T}_{\delta}$.

---

[2]We define a program as a string of bits, and we assume that the elements in $\mathcal{X}$ or $\mathcal{Y}$ can be encoded as bit-strings, i.e., $\mathcal{P} = \{0, 1\}^*$ and $\mathcal{X}, \mathcal{Y} \subseteq \{0, 1\}^*$, where $^*$ denotes the Kleene star operator.

[3]We additionally consider the tokenizer for a language model to be part of the bit-string $\theta$.

**Definition 3.6.** *Let $\theta$ be the bit-string encoding the weights of a pre-trained model and $\mathtt{T}$ be a task. We say model $\theta$ is $(b, \delta)$-adaptable to task $\mathtt{T}$ if:*

$$\mathrm{C}(\mathtt{T}_\delta \mid \theta) \leq b \tag{5}$$

Given this definition, we can now define the SAH as follows:

**Definition 3.7.** *(Informal) Let $\theta$ be a pre-trained model, $b$ a relatively small program size, and $\delta$ a relatively high performance. The **superficial alignment hypothesis** argues that there exist many complex tasks $\mathtt{T}$ (with high $\mathrm{C}(\mathtt{T}_\delta)$) of practical interest, to which model $\theta$ is $(b, \delta)$-adaptable.*

This grounds our intuitions that we can easily adapt a pre-trained language model for many tasks, leveraging the information already present in $\theta$. However, our definition has one informal point: what are *relatively low $b$* and *relatively high $\delta$*? Is a program of $1k$ bits and $90\%$ accuracy superficial? Since there does not seem to be a principled way of defining a single budget–performance value $(b, \delta)$ that we would call 'superficial', we study the tradeoff between these parameters and abstain from choosing a single target value to analyse.

### 3.3. Connection to previous work

Our notion of Task Complexity is inspired by classical ideas from algorithmic information theory. In particular, Kolmogorov complexity (Kolmogorov, 1965) measures the information content of a finite string $y$ as the length of the shortest program[4] that outputs $y$:

$$\mathrm{C}(y) \overset{\text{def}}{=} \min_{\mathtt{P} \in \mathcal{P}} \{\mathrm{len}(\mathtt{P}) : \mathtt{P}() = y\} \tag{6}$$

Kolmogorov complexity, however, is generally not well suited to study the complexity of machine learning (ML) tasks. First, a task $\mathtt{T}$ includes a distribution $p$ over a potentially infinite set of inputs $\mathcal{X}$, rather than being defined as a single finite string. Second, ML is inherently concerned with approximate solutions: achieving $90\%$ accuracy (or even $99\%$) in a task $\mathtt{T}$ may require far less information than achieving $100\%$, yet Kolmogorov complexity does not naturally express this performance-complexity tradeoff.

Algorithmic rate-distortion theory (Vereshchagin & Vitányi, 2010) addresses the second limitation above by allowing approximate outputs, defining complexity as the length of the shortest program which produces an output within a given distortion level of a target value. Interpreting this target to be an input $x$, and the distortion to be measured via a scoring function $\mathcal{S}$, we get:

$$\mathrm{C}(x_\delta) \overset{\text{def}}{=} \min_{\mathtt{P} \in \mathcal{P}} \{\mathrm{len}(\mathtt{P}) : \mathcal{S}(x, \mathtt{P}()) \geq \delta\} \tag{7}$$

---

[4]Note that, in this definition, the program takes no input value.

As can be seen, this equation measures the complexity of representing a fixed input $x$ with a specific accuracy, and it thus does not address the first limitation above. Task complexity adapts Equation (7) by replacing this single-target distortion with a 'distortion' metric defined over the entire input space $\mathcal{X}$. We thus say that task complexity generalises these two notions of complexity, being more suited to the study of machine learning tasks (see Remark A.5).

Task complexity is also related to the minimum description length (MDL) principle (Grünwald, 2007) and, more specifically, to prior probing methods inspired by it (Voita & Titov, 2020; Pimentel et al., 2020a; Pimentel & Cotterell, 2021). These methods measure how compactly a dataset can be compressed given a pretrained model, and are typically used to analyse how easy-to-access is some information encoded in the model's representation.[5] In contrast, task complexity measures the length of the entire adaptation program required to achieve a target performance on the task. Also, by measuring program complexity rather than data description length alone, our framework lets us compare fundamentally different views of the SAH (e.g., inference-time control vs. data driven fine-tuning) within a single unified metric.

## 4. Estimating Task Complexity

We wish to compute the $(b, \delta)$-adaptability of pre-trained models $\theta$ to common tasks $\mathtt{T}$. However, the conditional task complexity $\mathrm{C}(\mathtt{T}_\delta \mid \theta)$, on which adaptability depends, is uncomputable.[6] Hence, we upper bound this conditional complexity by finding programs $\mathtt{P}$ which adapt model $\theta$ to perform a task; after measuring this program's performance $\delta$, we know that its length will naturally upper bound the task's complexity $\mathrm{C}(\mathtt{T}_\delta \mid \theta) \leq \mathrm{len}(\mathtt{P})$.

Our definitions of program $\mathtt{P}$, and how to measure its length $\mathrm{len}(\mathtt{P})$ depends on what we define as our machine $\mathrm{M}$. Task complexity, though, is invariant (up to an additive constant) to this choice of machine $\mathrm{M}$, as long as it is Turing complete (Theorem A.4). Here, we aim for a definition of $\mathrm{M}$ that closely matches the programs that are used in practice to adapt pre-trained models for various tasks. Hence, we define $\mathrm{M}$ to be a python interpreter, where all python libraries are considered part of the machinery of the system. This means that we do not consider the code of libraries (such as `transformers` or `numpy`) to be part of the length of a program $\mathtt{P}$. Any real program designed to adapt and run

---

[5]Note that, due to the injectivity of transformers with real-valued hidden representations outputs (Sutter et al., 2025; Nikolaou et al., 2026), an LLM's representation always preserve all the information encoded in its inputs (Pimentel et al., 2020b). For this reason, prior work has focused on "easy-to-access information", as opposed to the total information amount.

[6]Our definition extends Kolmogorov complexity which is uncomputable (see Corollary A.6).

```
from transformers import ...

compressed_data = [0.21, 1.52, ...]

for batch in compressed_data:
  batch = decompress(batch, model)
  model(batch).loss.backward()
  optimizer.step()

output = model(eval_input)
```

*(a)* Data methods

```
from transformers import ...

adapter_weights = [
    0.3425,
    2.5145,
    ...
]
model.add_adapter(adapter_weights)

output = model(eval_input)
```

*(b)* Parametric methods

```
from transformers import ...

compressed_prompt = 0.124512213415
prompt = decompress(
    compressed_prompt,
    model
)

full_input = prompt + eval_input
output = model(full_input)
```

*(c)* Inference-control methods

*Figure 2.* Pseudo-code of the programs P constructed by strategies in Section 4. Each program includes its compressed data or parameters, and its size is usually dominated by such terms (Figure 10). We explain the details of how we measure program size in our experiments in Appendix B.

language models can use these libraries, without the need to re-write all the utilities defined by these. To measure the length of a program $\text{len}(\text{P})$, we need to measure the length in bits of the python script, and any of the model weights and data used in the script, like illustrated in Figure 2. As an illustrative example, if we have a python script that loads a pre-trained model $\theta$, and some data $\mathcal{D}$ that is used to fine-tune the model $\theta$, we would measure the size of the script in bits, plus the size of $\mathcal{D}$ in bits. In Appendix B we leave additional details regarding the measurements of $\text{len}(\text{P})$.

To achieve a tight upper bound on $\text{C}(\text{T}_\delta \mid \theta) \leq \text{len}(\text{P})$, we should look for short programs which we expect to achieve high performance. How do we select such programs? We do that by leveraging the connection between our definition of task complexity and the three views of superficiality reviewed in Section 2. Each of these views represents different strategies for how to construct short, high-performance programs to adapt pre-trained models. These strategies typically leverage training data $\mathcal{D}$ drawn from task $\text{T}$, and generate programs $\text{P}$ that adapt the model to perform well on $\text{T}$. Importantly, we only measure the length of the final program $\text{P}$ output by these strategies, and not the size of the procedure generating it.

Together, the length ($b$) and performance ($\delta$) of these programs $\text{P}$ allow us to estimate the points on a *Pareto-optimal* length–performance $(b, \delta)$ curve, which will constitute our adaptability estimates. We now describe three procedures for constructing programs, one for each view from Section 2.

① **Data methods.** This procedure assumes access to a dataset $\mathcal{D}$ available for this task, and the pre-trained model $\theta$. It then draws a subset $\mathcal{D}' \subseteq \mathcal{D}$ of the data and trains the model on this data. Before performing a gradient step on a target batch, however, it uses the model $\theta$ to compress this batch via arithmetic coding (Deletang et al., 2024) and writes this compressed data $\mathcal{D}'_c$ into program $\text{P}$. Program $\text{P}$ then iteratively uses the model $\theta$ to decompress the data, and to update the model, being able to reconstruct such a fine-tuning procedure. Finally, the program then uses

this fine-tuned model to perform inference on evaluation examples. The pseudo-code for this resulting program $\text{P}$ is shown in Figure 2a. The length of program $\text{P}$ then mostly depends on the size of the compressed data $\text{len}(\mathcal{D}'_c)$, with a constant overhead for, e.g., the size of the code used to decompress and train on this data. We term this procedure as **Subset Training**. The special case where the full dataset is used is labeled here as **Full Dataset**, which we use as a baseline for this approach.

② **Parametric methods.** This procedure again assumes access to a dataset $\mathcal{D}$ and the pre-trained model $\theta$. It then first randomly initialises the weights of an adapter $\theta_{\text{ADAPT}}$ for model $\theta$. These adapter weights $\theta_{\text{ADAPT}}$ are then trained using the data in $\mathcal{D}$, and written into a program $\text{P}$. The pseudo-code for these programs $\text{P}$ is shown in Figure 2b. Notably, the program $\text{P}$ directly encodes the adapter weights, which it uses to perform inference over the evaluation data. The program $\text{P}$'s length thus mostly depends on the size of these adapter parameters $\theta_{\text{ADAPT}}$, with a constant overhead for the size of the code to apply the adapter and run the model. We use two variations of this procedure, which differ in how to design and train the adapter $\theta_{\text{ADAPT}}$. The first, **LoRA** (Hu et al., 2022), uses a number of low-rank adapter layers when fine-tuning a model. The second, **Bayesian-LoRA** (Meo et al., 2024), is similar, but directly optimises the ranks (number of parameters) and quantisation levels (precision of each parameter) in each LoRA layer, allowing a model to be adapted with fewer bits. We also consider adapting the full model as a baseline, which we label as **Full Model**.

③ **Inference-control methods.** As before, this procedure assumes access to dataset $\mathcal{D}$ and model $\theta$. The procedure then draws a very small set of samples $\mathcal{D}' \subseteq \mathcal{D}$ and combines them into a prompt $\kappa$, which it compresses into $\kappa_c$ using arithmetic coding (using model $\theta$) and writes into a program $\text{P}$. Program $\text{P}$ then decodes $\kappa$ and adds it into the context of model $\theta$ before running inference on the evaluation examples. The pseudo-code for program $\text{P}$ is shown in Figure 2c. For this strategy, the length of program $\text{P}$ mostly

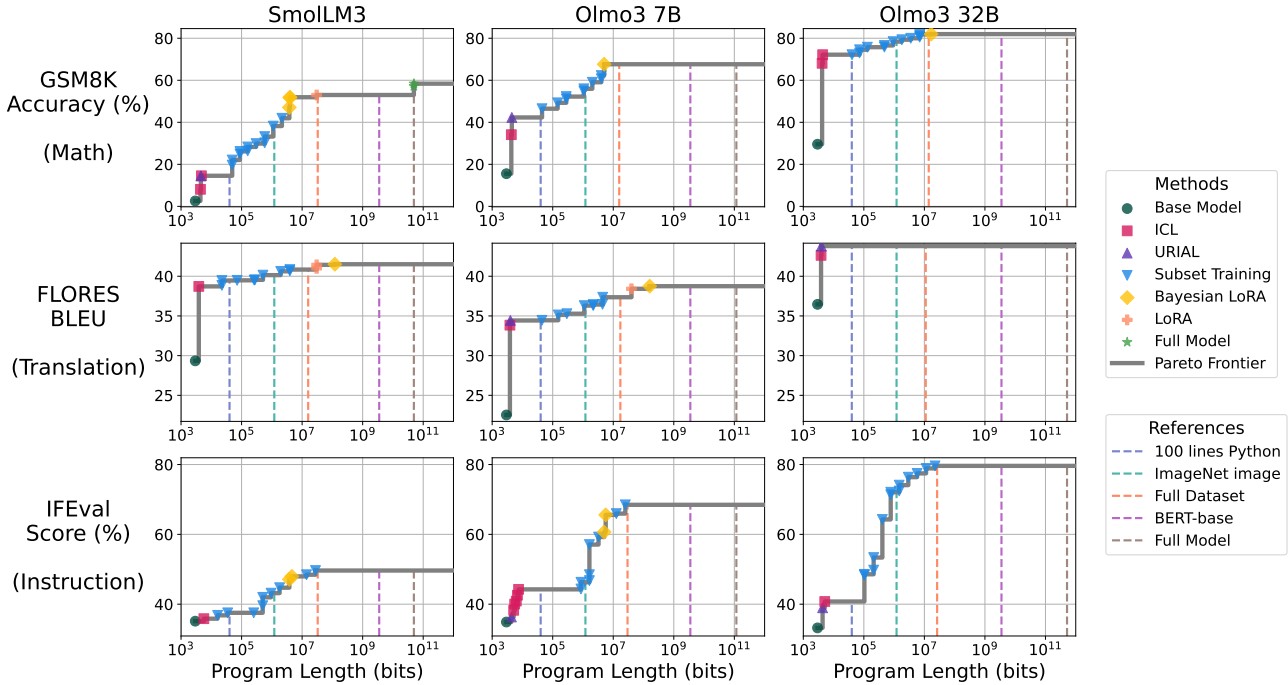

*Figure 3.* The program length vs. performance Pareto curves for SmolLM3 and Olmo3 in our three analysed tasks: mathematical reasoning, machine translation, and instruction following. We test each method described in Section 4, but show only the optimal lengths and performances. To provide some intuition about these program lengths, we also mark the size in bits of more commonly known references.

depends on the size of compressed prompt $\text{len}(\kappa_c)$, with a constant overhead for the the size of the code to decompress the prompt and run the model. We experiment with two variations of this procedure. First, **In-Context Learning** (ICL; Brown et al., 2020), where the prompt $\kappa$ contains examples from $\mathcal{D}'$. Second, **URIAL** (Lin et al., 2024), where an explanation of the task $\text{T}$ is added to prompt $\kappa$ in addition to the examples in $\mathcal{D}'$. Finally, we also use an empty prompt $\kappa$ as a baseline, which we label as **Base Model**.

## 5. Experiments

Our experiments estimate the length–performance tradeoff of the $(b, \delta)$-adaptability of real models $\theta$ on real tasks $\text{T}$.

### 5.1. Experimental setup

**Models.** As our models $\theta$, we use SmolLM3 3B (Bakouch et al., 2025), Olmo3 7B, and Olmo3 32B (Olmo et al., 2025). These models provide a wide coverage of model size and are open-source, providing not only weights but also their training data and code. For most of our experiments, we will use as our pre-trained models $\theta$, more specifically, the last checkpoint of these models' first training stage, i.e., the last checkpoint at `stage-1` of the SmolLM3 and the Olmo3 training pipelines, respectively.

**Tasks.** We select three diverse NLP tasks for our experiments, which have been extensively studied in prior work: Mathematical reasoning (GSM8K; Cobbe et al., 2021),[7] English to French machine translation (FLORES-200; Costa-Jussà et al., 2024), and instruction following (IFEval; Zhou et al., 2023b). We evaluate performance using accuracy, BLEU, and verified success rate, respectively. Additional details about these tasks are provided in Appendix C.

**Hyperparameters.** For every method described in Section 4, we perform a hyperparameter sweep to obtain a set of length–performance points $(b, \delta)$. Based on these points, we construct a Pareto optimal curve by selecting all the $(b, \delta)$ pairs which are not dominated by any other, i.e., pairs for which no other pair performs at least as well in terms of both length and accuracy. We provide a more detailed description of these hyperparameter sweeps in Appendix D.

## 6. Results

Figure 3 shows our estimates of the Pareto curve between program length and task performance $(b, \delta)$ for the three analysed tasks and models. In the following, we discuss our main findings.

---

[7]We draw training examples from MetaMath (Yu et al., 2024), which extends the training set of GSM8K.

## 6.1. Evidence for superficial alignment to complex tasks

It is possible to adapt pre-trained LLMs to achieve high performances using surprisingly short programs. On GSM8K (first row of Figure 3), all pre-trained models achieve low accuracy between 2.6% and 29.6%. However, a program of only 4358 bits is sufficient to increase OLMo3-32B's accuracy to 72.2%. Similar trends hold for machine translation (second row) and instruction following (third row). The pre-trained OLMo3-7B achieves 22.63 BLEU score on English to French translation, but can be adapted to achieve 34.43 BLEU with a program of only 3992 bits. On IFEval, maximum performance is achieved with all models with programs only slightly longer than $10^7$ bits (1.25MB). While these pre-trained models used terabytes of data during pre-training (see Appendix E), adapting them to achieve high performance on these tasks can be done using only megabytes of information. We interpret these results as evidence in favour of the superficial alignment hypothesis.

## 6.2. Different views of superficiality inform different regions of the Pareto curve

The estimated $(b, \delta)$ Pareto curves in Figure 3 contain points from various methods corresponding to the different views of superficiality described in Section 2. Figure 1 shows how each of these views contributes to a particular region of the Pareto curve (plots for other models and tasks are provided in Figure 6). At the shortest program lengths, we see that the Pareto curve is mostly informed by the ③ inference-control view. These methods (URIAL and ICL) bring modest gains in performance with minimal program size. In the medium regimes of program lengths, we observe that the ① data view dominates. This method (Subset Training) improves performance over the the inference-control methods, but at the cost of encoding much more data in the program P. Finally, the longest programs always correspond to the ② parameter view (LoRA and Bayesian-LoRA), whose methods directly encode parameters in P. While the three views supporting the SAH (①, ② and ③) seemed orthogonal at first, they are now thus directly compared on a common, quantitative footing. Instead of competing explanations, they are rather important and complementary techniques to estimate the length–performance $(b, \delta)$ Pareto curves.

## 6.3. Revisiting previous claims about superficiality

In light of the results discussed above, we now revisit previous claims made about the superficiality of LLM adaptation.

**Raghavendra et al. (2024): Fine-tuning on few samples does not saturate performance.** The original formulation of the SAH left unclear what 'knowledge' and 'capabilities' implied. Both Raghavendra et al. (2024) and Lambert (2025)

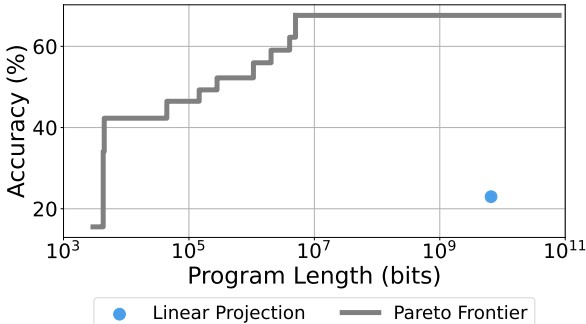

*Figure 4.* Performance of the linear projection fine-tuning proposed by Chen et al. (2025) in comparison to our estimated Pareto curve. The performance of the method is far from the Pareto curve.

interpret it as implying that fine-tuning on few data points should quickly saturate performance on a task; they then found that this is generally not the case. We intentionally incorporated the target performance level $\delta$ in our definition of SAH (Definition 3.6), as some tasks may require arbitrarily large amounts of information to achieve particularly high performance levels. For example, Olmo3 32B can be adapted to achieve 78.4% in GSM8K with a program as small as a single ImageNet image. However, we are not able to substantially improve upon these results even in our largest programs, suggesting that perhaps considerably more information is needed in order to saturate the task.

**Liu et al. (2024): Fine-tuning updates can be worth as little as 1 bit per parameter.** There is a growing interest in compressing the weight difference between pre-trained and fine-tuned model, which is sometimes termed delta compression (Ping et al., 2024; Huang et al., 2025; Yadav et al., 2025). They are similarly motivated by the intuition that fine-tuning adds less information to a model than pre-training, and therefore it may be possible to compress this update. Liu et al. (2024), for example, compress the difference between a pre-trained and fine-tuned model to approximately 1 bit per parameter, with minimal performance impacts on their analysed tasks. They interpret these results as indicative of the low amount of information gained during fine-tuning. Under our framework, these results understate the amount of information that fine-tuning adds to a model. For example, the size of a full fine-tuning dataset is much smaller than the size of 1-bit per parameter of an LLM. Comparing our smallest model (SmolLM3) to our largest dataset (GSM8K), the dataset size is $3 \times 10^7$ bits while the model has $3 \times 10^9$ parameters. These methods, thus, would produce programs P that are neither small nor performant enough to be Pareto optimal under our operationalisations.

**Chen et al. (2025): Fine-tuning on reasoning tasks is not superficial.** Chen et al. (2025) critique the SAH for not rigorously defining what 'superficial knowledge' means. They then define superficial knowledge as the performance a pre-trained model can achieve when only adapting the linear projection of the output layer. This is based on the intuition that if learning a linear transformation on top of a pre-trained model is sufficient for performing well on a task, then most of the knowledge required for the task is already present in the pre-trained model itself. Empirically, they show that reasoning tasks such as GSM8K require more than superficial knowledge, because fine-tuning only the output layer is not sufficient for achieving good performance. Our framework reveals that their approach might not be optimal for finding short adaptation programs. In Figure 4, we show the length–performance of their method compared to the Pareto curve when adapting OLMo3-7B to GSM8K. This figure illustrates that training only the output layer leads to neither short programs nor high accuracies (see similar results for other tasks in Figure 7 in Appendix F). More broadly, Chen et al.'s approach is closely related to probing, and—as argued by prior work in that field (Hewitt & Liang, 2019; Pimentel et al., 2020b)—we see no reason why 'superficiality' should be restricted to linearly encoded knowledge.

## 7. How Does Pre- and Post-training Affect the Superficiality of Adaptation?

Finally, to demonstrate the usefulness of our definitions beyond revisiting previous claims about superficiality, we now show that $(b, \delta)$ Pareto curves can shed light on how post-training affects the superficiality of adaptation.

**Experimental setup.** We compute $(b, \delta)$ Pareto curves for mathematical reasoning (GSM8K) and instruction following (IFEval) using randomly initialised models as well as the final pre- and post-training[8] checkpoints of SmolLM3 3B and Olmo3 7B (results for two additional tasks are available in Figure 8 of Appendix F). We use the same evaluation setup, including the same minimal prompting templates, for all these checkpoints. Figure 5 shows the Pareto curves obtained when running our methods on these models $\theta$.

**Pre-training makes strong performance accessible.** As expected, in this figure, we see that pre-training enables the model to reach much higher task performance compared to randomly initialised checkpoints. For example, in the case of Olmo3-7B, the best performance we are able to achieve on mathematical reasoning (GSM8K) improves from 1.1% to 67.6%, and on instruction following (IFEval) from 26.6% to 68.5%. Further, this figure also shows that the allowed

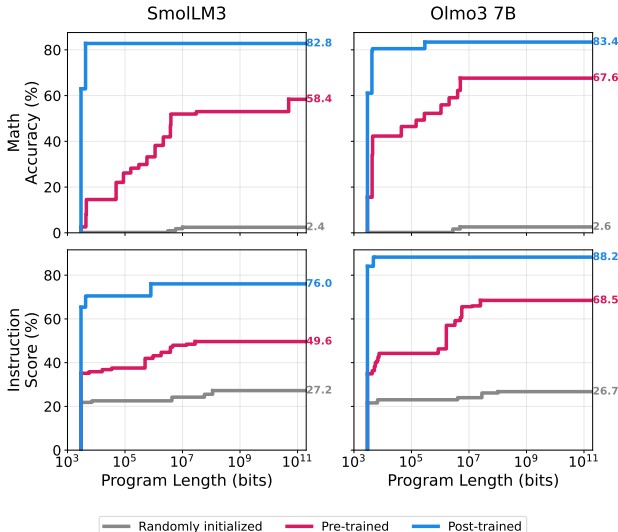

*Figure 5.* Estimated complexity curves conditioned on pre- and post-trained checkpoints of SmolLM3 and Olmo3 for math (GSM8K) and instruction (IFEval) tasks. Pre-training allows access of strong performance in these tasks. Post-training improves these performances, but more importantly, it greatly reduces the complexity to reach such performance.

program size significantly affects the performance of pre-trained checkpoints. For example, with programs of $10^5$ bits, the performance obtained in each setting is still far below its maximum performance. Finally, at the maximum achieved performance levels $\delta$, the task complexity is relatively high; most evidently in SmolLM3 on GSM8K, performance only plateaus at $5 \times 10^{10}$ bits (around 6.2 GB).

**Post-training collapses the complexity of achieving strong performance.** As expected, post-training further increases task performance, e.g., from 58.4% to 82.8% in mathematical reasoning and from 49.6% to 70.5% for instruction following for SmolLM3.[9] More importantly, however, post-training collapses the complexity of achieving the maximum performance. For example, for pre-trained Olmo3-7B the complexity of achieving its maximum performance is $5 \times 10^6$ bits for GSM8K, and $2.5 \times 10^7$ bits for IFEval. In contrast, for post-trained models, program size has minimal effect on performance beyond a size of $10^4$ bits. These results provide an information-theoretic interpretation of what it means for post-training to resurface what was learned in pre-training: the complexity of accessing strong performance is substantially reduced.

---

[8]See Appendix E for details on the data used for pre- and post-training these models.

[9]We note that the training data for both GSM8K and IFEval were included in the post-training of both SmolLM3 and Olmo3 (see Appendix E), which could partially explain this effect. We additionally experiment on FLORES and e-SNLI (Camburu et al., 2018) to decouple this effect (Figure 8).

# 8. Conclusion

The superficial alignment hypothesis has lacked a precise definition, which led both to seemingly orthogonal interpretations of how to support this hypothesis and to important critiques against it. We address this here by defining task complexity—the length of the shortest program that achieves a target performance on a task—which we use to formalise the SAH as claiming that access to a pre-trained model drastically reduces the complexity of achieving high performance on many tasks that would otherwise be highly complex. Our framework unifies previous arguments in support of the SAH (the data view, the parametric view, and the inference control view, see Section 2) as proposing different strategies for finding short, high-performance programs for a target task. Experimentally, we estimate the task complexity of mathematical reasoning, machine translation, and instruction following, conditioning these complexities on three different models. We find that programs as small as a few kilobytes can already adapt pre-trained models to achieve strong performance on these tasks. By analysing intermediate training checkpoints of LLMs, we show that pre-training enables access to strong performance, though often at high complexity (requiring megabytes to gigabytes long programs), while post-training dramatically collapses this complexity, making high performances accessible with less than a kilobyte. We believe this to be an important step towards understanding the concept of superficiality in LLMs, and the information contained in them.

# 9. Limitations and Future Work

Our operationalisation of the SAH inherits fundamental properties of algorithmic information theory that shape and limit what claims we can make. First, because task complexity is uncomputable (Corollary A.6), our empirical estimates provide only upper bounds. Therefore, our $(b, \delta)$ Pareto curves show which performance levels are achievable with different program lengths, but they are not provably tight, as there may exist shorter programs we have not discovered. Second, while we can reasonably upper-bound the conditional task complexity $C(T_\delta \mid \theta)$ using the three views on superficiality, estimating the task complexity $C(T_\delta)$ is difficult, as it requires finding the shortest program to solve $T_\delta$ without access to any pre-trained model. This prevents us from directly measuring the information $I(T_\delta; \theta)$ (see Definition 3.5), which is why our formalisation of the SAH (Definition 3.7) relies solely on conditional complexity. We hope future work can explore the tightness of our approximations.

Beyond the SAH, task complexity may offer useful perspectives on other problems in ML. For instance, work on quantifying the safety of open-weight model releases (Wallace et al., 2025; Che et al., 2025) deals with measuring how easily unsafe capabilities can be accessed. Our framework

naturally operationalises this as $I(T_\delta; \theta)$ for some unsafe task $T$ and performance $\delta$: if low, releasing the model weights does not make $T_\delta$ easier to solve.

# Impact Statement

This paper presents work whose goal is to advance the field of machine learning. There are many potential societal consequences of our work, none of which we feel must be specifically highlighted here.

## Acknowledgements

We would like to thank Michael Rizvi-Martel, Verna Dankers, Nicholas Meade, Benno Krojer and Sebastian Bordt for important discussions and feedback during this project. We would also like to thank Philip Whittington and Konrad Szafer for feedback on this manuscript. This research was enabled in part by compute resources provided by Calcul Québec (calculquebec.ca), the Digital Research Alliance of Canada (alliancecan.ca), and software and technical help provided by Mila (mila.quebec). IT acknowledges support from Dutch National Science Foundation (NWO Vici grant VI.C.212.053) and IVADO.

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

## A. Theoretical Properties of Task Complexity

The definition of Kolmogorov complexity is commonly attributed to Kolmogorov (1965), but was independently described by Solomonoff (1964) and Chaitin (1966), and is given in Equation (6). In this section, we list a number of properties of task complexity which are closely related to the notion of Kolmogorov complexity. First, we present three useful bounds on conditional and unconditional task complexity.

**Remark A.1.** *For any task* $T$ *and performance* $\delta$, *conditioning on a model* $\theta$ *can only reduce task complexity:*

$$C(T_\delta \mid \theta) \leq C(T_\delta) \tag{8}$$

*Proof.* Let $P^\star$ be a shortest (input-free) program achieving $\texttt{score}_T(P^\star) \geq \delta$, so $C(T_\delta) = \texttt{len}(P^\star)$. We can augment this program $P^\star$ with an ignored input $\theta$, and it would still solve task $T$ at performance $\delta$.[10] $\qquad\square$

**Remark A.2.** *For any task* $T$, *performance* $\delta$ *and model* $\theta$, *(unconditional) task complexity is upper bounded by the task complexity conditioned on* $\theta$ *plus an overhead for encoding* $\theta$:

$$C(T_\delta) \leq C(T_\delta \mid \theta) + K(\theta) + O(1) \tag{9}$$

*Proof.* Let $P^\star$ be a shortest program which, conditioned on $\theta$, solves $T_\delta$, i.e., $\texttt{score}_T(P^\star_\theta) \geq \delta$. We thus have that $C(T_\delta \mid \theta) = \texttt{len}(P^\star)$. We can construct a program $P'$ that directly encodes $\theta$ (which requires $K(\theta)$ bits), which it uses to execute $P^\star$ (requiring $\texttt{len}(P^\star)$ bits plus an $O(1)$ overhead). Thus:

$$C(T_\delta) \leq \texttt{len}(P') = K(\theta) + \texttt{len}(P^\star) + O(1) \tag{10a}$$
$$= K(\theta) + C(T_\delta \mid \theta) + O(1) \tag{10b}$$

$\qquad\square$

**Remark A.3.** *Task complexity is monotonic on the performance* $\delta$, *meaning that, for any task* $T$ *and performances* $\delta_1 \leq \delta_2$:

$$C(T_{\delta 1}) \leq C(T_{\delta 2}) \tag{11}$$

*with the same holding for conditional task complexity.*

*Proof.* Any program $P$ that achieves performance $\delta_2$ also achieves performance $\delta_1$. Its length is therefore an upper bound on $C(T_{\delta 1})$. $\qquad\square$

**Theorem A.4.** *Task complexity is invariant up to an additive constant to the choice of universal Turing Machine* $M$. *In other words, for any two universal Turing Machines* $M_1$ *and* $M_2$, *there exists a value* $\alpha$ *such that for any task* $T$ *and performance* $\delta$:

$$|C_{M_1}(T_\delta) - C_{M_2}(T_\delta)| \leq \alpha \tag{12}$$

*where* $C_M$ *is the task complexity using machine* $M$, *and* $\alpha$ *depends exclusively on the machines, but not on the task.*

*Proof.* We first show that $C_{M_1}(T_\delta) \leq C_{M_2}(T_\delta) + \alpha_{1,2}$ for some $\alpha_{1,2}$ depending only on $M_1$ and $M_2$. As $M_1$ is a universal turing machine, there exists a program $P_{sim}$ that simulates $M_2$ on $M_1$. That is, for any program $P$ and input $x$:

$$M_1(P_{sim}, \langle P, x \rangle) = M_2(P, x), \tag{13}$$

where $\langle P, x \rangle = \bar{P} x$, with $\bar{P}$ a self-delimiting encoding of $P$, so that both $\langle \cdot, \cdot \rangle$ and its inverse are computable (Li & Vitányi, 2008).

Now, let $P^\star$ be a shortest program solving $T_\delta$ on $M_2$, so $\texttt{len}(P^\star) = C_{M_2}(T_\delta)$. We construct a program $P'$ for $M_1$ that, on input $x$, forms $\langle P^\star, x \rangle$ and runs $P_{sim}$ on it. By construction, $P'$ computes the same outputs as $P^\star$ does on $M_2$, so it solves

---

[10]Depending on the UTM formalisation, ignoring extra inputs may require an additional $O(1)$ term in the inequality.

$T_\delta$. Its length is $\text{len}(P_{sim}) + \text{len}(P^\star) + O(1)$, where the $O(1)$ term accounts for the encoding. Since this upper bounds $C_{M_1}(T_\delta)$, we obtain:

$$C_{M_1}(T_\delta) \leq C_{M_2}(T_\delta) + \alpha_{1,2} \tag{14}$$

with $\alpha_{1,2} = \text{len}(P_{sim}) + O(1)$.

By symmetry of $M_1$ and $M_2$, we also have $C_{M_2}(T_\delta) \leq C_{M_1}(T_\delta) + \alpha_{2,1}$. Setting $\alpha = \max(\alpha_{1,2}, \alpha_{2,1})$ gives the result. □

We now show that task complexity generalises Kolmogorov complexity, having it as a special case.

**Remark A.5.** *Task complexity generalises Kolmogorov complexity.*

*Proof.* Let $s$ be any finite string. We write its Kolmogorov complexity as $C(s)$. Now, let $\delta = 1$ and construct a task $T$ with no inputs, and whose goal is solely to output this string $s$. Formally: $\mathcal{X} = \{\varepsilon\}$, $\mathcal{Y} = \{0, 1\}^*$, $p(\varepsilon) = 1$, and $\mathcal{S}(x, y)$ returns 1 if and only if $y = s$ and 0 otherwise. Then we have exactly: $C(T_\delta) = C(s)$. □

As task complexity reduces to Kolmogorov complexity in a special case, it is easy to show it is both uncomputable and unbounded. Uncomputability means there does not exist a program such that, for all tasks $T$ and all performances $\delta$, it computes $C(T_\delta)$. Unboundedness means that there does not exist any constant $k$ which is larger than all task complexities.

**Corollary A.6.** *Task complexity is uncomputable.*

**Corollary A.7.** *For any $k \in \mathbb{N}$, there exists a task $T$ and performance $\delta$ such that $C(T_\delta) \geq k$.*

# B. How to compute $\text{len}(P)$

Our definitions in Section 3 rely on a Universal Turing Machine $M$ and binary-string programs $P$; the size of program $P$ is then simply its length in bits. In practice, our operationalisation into experiments (Section 5) relies on `python` being the machine $M$ and on multiple files representing parts of the program and specific information (about the program's data or adapted parameters) which it loads. For both the data and inference-control methods, thus, our program $P$'s length is thus the size in bits of the actually used Python script, plus the size of a separate file which stores its required data. For the parameter methods, our program $P$'s length is the size in bits of the used Python script, plus the size of a separate file which stores the saved adapter parameters. The implementation of all of our measurements are in our repository.

**The data file size.** We encode any data used in our methods via arithmetic coding with the model $\theta$, following Deletang et al. (2024). This is an efficient method to losslessly compress and decompress data if given access to an LLM. To speed up our experiments, we estimate the size of this encoded data as the negative log-likelihood of the forward pass of model $\theta$ on the data, as similarly done by prior work measuring the size in bits of data (Morris et al., 2025; Pezeshkpour, 2023).

**The weights file size.** We always train our models using 16 bit accuracy, and therefore we compute the size of the weights as 16 times the number of trainable parameters of the adapter. For Bayesian-LoRA, we compute the size of each adapter matrix as $q \times (r \times (d_{in} + d_{out}))$ where $q$ is the learned quantisation level, $r$ is the learned rank, and $d_{in}$ and $d_{out}$ are the dimensions for the matrix being adapted. Note that $r \times (d_{in} + d_{out})$ is the number of trained adapter parameters.[11]

**The Python script file size.** Given some python file, e.g. a `main.py`, we can measure its storage size in bytes as `wc -c main.py`. However, this does not consider the size of the code to run any of its dependencies, e.g. `import transformers`. We will assume that any library (built-in or not) is part of the machine $M$, and therefore ignore its measurements. This rule could be abused, for example, by creating a library that contains all of the code for a program $P$ of our experiments. We do not fall into this behaviour. In our experiments, the most niche and specific python library which we assume access to is gptzip, which implements the encoding of data via arithmetic coding with LLMs. The methods Base Model, ICL, URIAL, and Subset Training, LoRA, Bayesian-LoRA, and Full Model have, respectively, sizes 2952, 3704, 3704, 5704, 2832, 8376, 2080 in bits. These contributions mostly affect Base Model, ICL, and URIAL Figure 10.

---

[11]An additional strategy for optimizing the size of the weights is to compress them with `zip`. We experimented with compressing SmolLM3, Olmo3 7B, and Olmo3 32B, getting reductions of 20.7%, 20.5%, and 20.6%, respectively. However, this resulted in an increase of wall clock time for our experiments by up to 2 hours for each compression. We excluded this strategy from our results.

# C. Tasks

We are motivated to select tasks that: (i) cover a diverse set of NLP domains, and (ii) plausibly have high complexity $C(T_\delta)$ for some levels of performance. We select the following three tasks. We note that, to quantify the performance $\texttt{score}_\texttt{T}(\texttt{P})$ in each task, we use sample averages over finite test sets as an approximation to the expectation over the task's distribution $p$.

**Mathematical Reasoning.**   For this task, we use the GSM8K dataset (Cobbe et al., 2021). This dataset contains grade-school math word problems requiring multi-step arithmetic reasoning. We use MetaMath (Yu et al., 2024), which is an extended version of the training set, containing 395k examples. We measure the performance as accuracy of the final response.

**Machine Translation.**   For this task, we use the English to French portion of the FLORES-200 dataset (Costa-Jussà et al., 2024). We draw the first 100k examples from the training set, and use BLEU (Papineni et al., 2002) as the performance metric.

**Instruction Following.**   For this task, we use the IFEval dataset (Zhou et al., 2023b). The dataset contains queries as *'Explain concept X'* augmented with easily verifiable rules as *'use only lowercase'* or *'do not use commas'*. We measure performance as the average success of following these rules, and we verify each rule using the official implementation of the dataset. We generate a training set of 37k examples, using the Tulu 3 data (Lambert et al., 2025), which we detail in the next section.

## C.1. Generating Instruction Following Data

The original work about the superficiality of LLM adaptation focused on adapting them to follow instructions, and therefore, we found it important to add this to our set pf analysed tasks. Evaluating instruction following behaviour is not necessarily an easy task, and previous work relied on the use of expensive human annotations or LLM judges (Zhou et al., 2023a; Hewitt et al., 2024; Lin et al., 2024). Rather, we opted for using a simpler and more reliable evaluation method: IFEval (Zhou et al., 2023b), which implements 'verifiable instructions'. However, IFEval does not provide a training set, which is the reason why we have to create our own synthetic data for training. We start with the training data for IFEval created in Tulu (Lambert et al., 2025), consisting of 15k examples formatted for IFEval with its verifiable rules. However, these datapoints are designed for reinforcement learning with verifiable rewards (RLVR), and therefore do not provide a reference on how to solve the instructions. We use a model trained via RLVR on this dataset, Olmo3 32B Thinking, to generate 10 candidate solutions for every datapoint in Tulu; we then filter these to keep only the correct ones, and remove duplicate answers. Our final training set contains 37k datapoints.

# D. Hyperparameters

In this section, we provide a short description of the parameters we perform a search over in our experiments, when trying to find Pareto optimal length–performance pairs $(b, \delta)$. The full results of our hyperparameter sweeep is in Figure 9. The hyperparameters used are:

**For ICL.**   We sweep over the number of examples $n$ used in the in-context learning experiments. We increase this number until we run into memory limits on a single 80GB GPU.

**For URIAL.**   We perform no sweep for this method. We run our experiments with a hand-designed prompt to explain the task, and we run this method using $n = 2$ examples of the data.

**For Subset Training.**   We again sweep this method over the number of examples $n$ used. We experiment with subsets of the data with sizes in powers of 2, starting with $n = 8$ and finishing with $n = 32768$ examples. We additionally experiment with fine-tuning the model using the full dataset. For each experiment, we also use learning rates $10^{-4}$ and $10^{-5}$.

**For LoRA.**   We use learning rates $10^{-4}$ and $10^{-5}$, always training with the full dataset.

**For B-LoRA.**   We use scale learning rates of $10^{-2}$ and $2 \times 10^{-3}$, training on the full dataset. We do one experiment with rank 1 for every LoRA layer and no rank pruning, and another starting with rank 2, but pruning the rank according to the optimisation objective of the method (Meo et al., 2024).

**For Full Model.** We use learning rates $10^{-5}$ and $10^{-6}$, training on the full dataset. Each parameter is stored at 16-bit precision.

## E. The Data used in Pre- and Post-training

Here, we detail the data used to pre- and post-train our analysed models.

**SmolLM3.** Bakouch et al. (2025) report using 8 trillion tokens during the `stage-1` of training (i.e., for pre-training). Assuming each token to be 4 bytes (enough to encode each vocabulary item of 128k in its tokenizer), this implies 32 terabytes of data for pre-training. Bakouch et al. also report using 3.2 trillion tokens during `stage-2`, `stage-3`, and the long context training stage (which we will here refer, jointly, to mid-training). This amounts to a total of 12.8 terabytes of data. Finally, Bakouch et al. report using a total of 37.5 billion unique training tokens for post-training, which amounts to 150 gigabytes of data, which was gathered by combining multiple post-training datasets. In this data (which they name the SmolTalk2 dataset), they include data of verifiable instruction following based on rules of IFEval, and also synthetic questions sourced from GSM8K training set.

**Olmo3 7B.** Olmo et al. (2025) report using 5.93 trillion tokens in the `stage-1` of training (i.e., for pre-training). This amounts to 23.7 terabytes of data for pre-training. Olmo et al. also report using a total of 150 billion tokens for `stage-2` and long context specialized training (i.e., for mid-training). This amounts to 600 gigabytes of data. Finally, Olmo et al. report using a total of 45 billion tokens in the post-training of the Olmo3 family. This amounts to 180 gigabytes of data. This includes stages of supervised fine-tuning, and also reinforcement learning with verifiable rewards. They release different variants of post-trained checkpoints, and we opt to use the last checkpoint in the `instruct` version.

# F. Extra Results

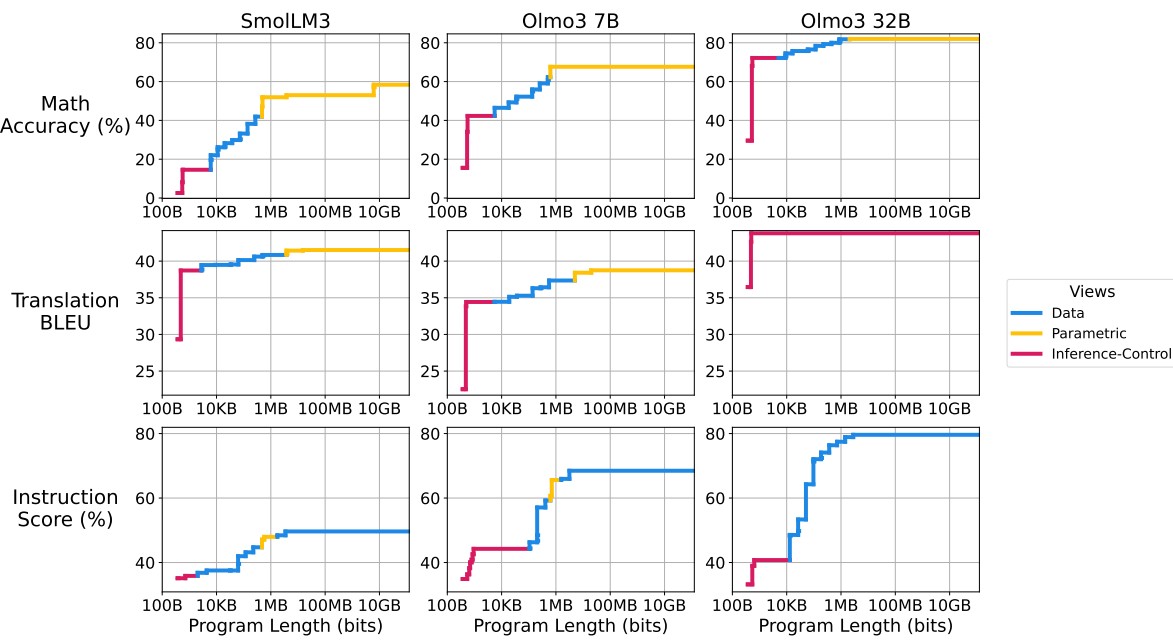

*Figure 6.* Length–performance Pareto curve of SmolLM3 3B, Olmo3 7B, and Olmo3 32B in Math (GSM8K), Translation (FLORES) and Instruction (IFEval). Different views of superficiality inform different regions of the Pareto curve, with some variance depending on the choice of model and task.

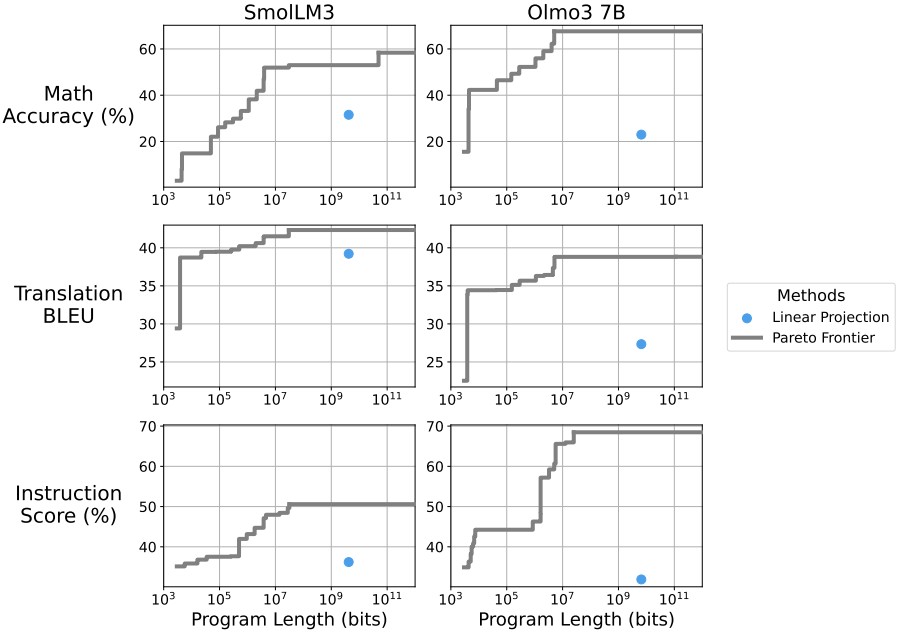

*Figure 7.* Performance of the linear (head-only) fine-tuning approach proposed by Chen et al. (2025) and our estimated Pareto curves in Math (GSM8K), Translation (FLORES) and Instruction (IFEval).

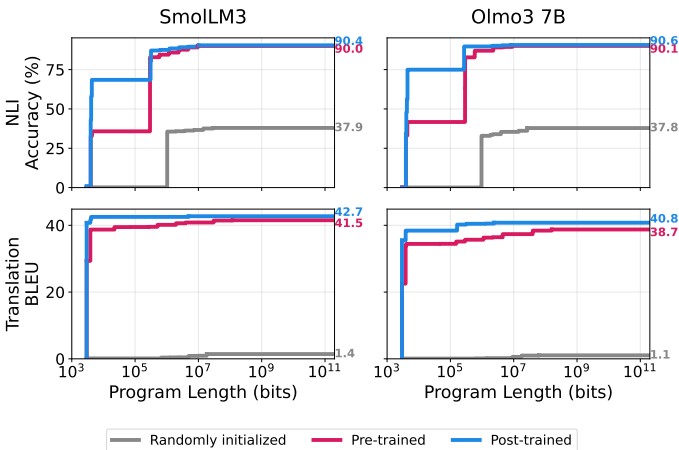

*Figure 8.* Estimated complexity curves conditioned on pre- and post-trained checkpoints of SmolLM3 and Olmo3 for FLORES and eSNLI.

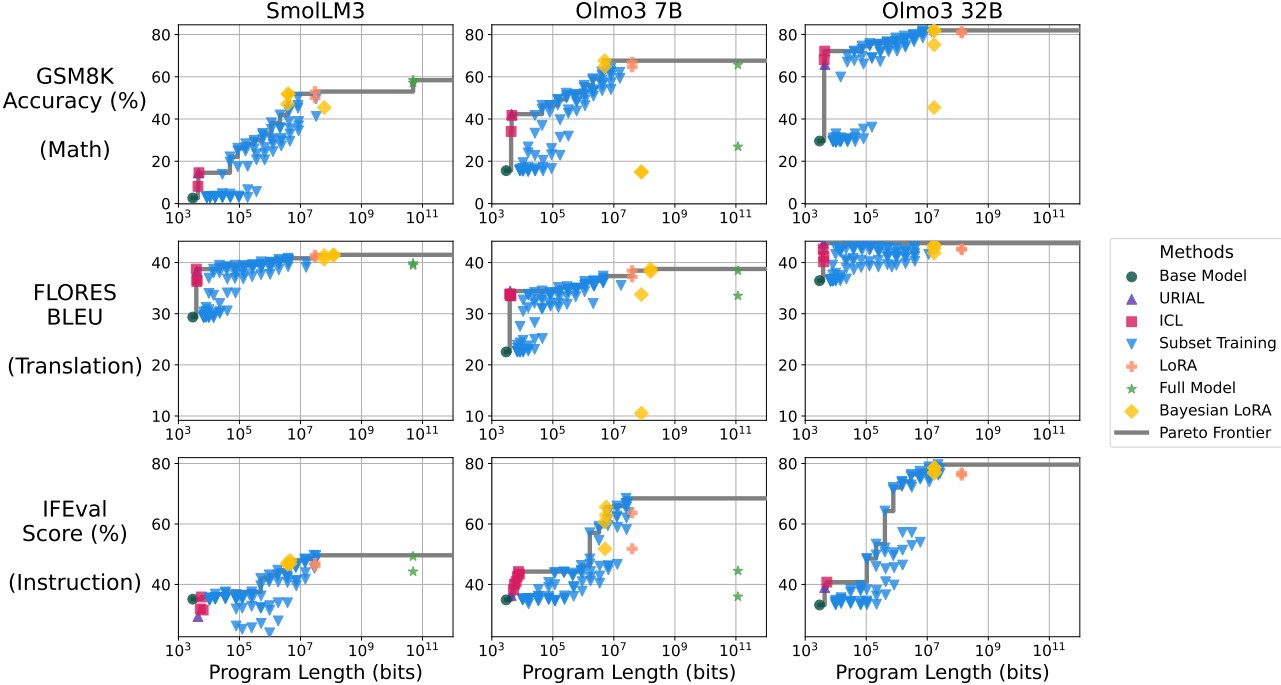

*Figure 9.* The program lengths vs. performance Pareto curves for SmolLM3 and Olmo3 in GSM8K and FLORES. We test each method described in Section 4, and show their message lengths and performances obtained. This is an extended version of Figure 3 with all the results from our hyperparameter sweep.

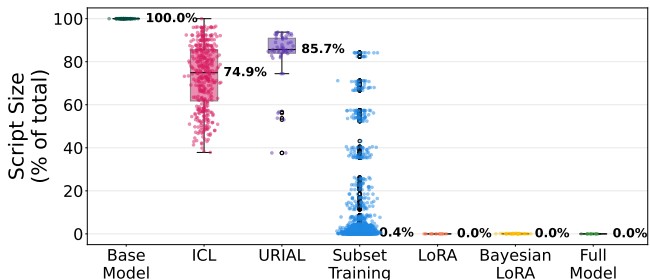

*Figure 10.* Distribution of percentage of the contribution to program length made by measuring the script size of each method described in Section 4 across all of our experiments. The script size mostly makes a substantial contribution in the Base Model, ICL and URIAL methods.

## G. Combining the Parametric and the Data View

In section Section 4, we showcased how every argument for superficiality can be unified under our task complexity lens. Each argument (the data view, the parametric view, and the inference-control view) proposes different strategies with which to build a program $P$ with small length to adapt a model for a task. Since these strategies are roughly orthogonal, we hypothesise that we can compose them together to achieve tighter Pareto curves of $(b, \delta)$ values. Here, we develop a method to combine the ② parametric view with the ① data view, in which the program $P$ needs to encode only few batches of the task, and few parameters which will be used to decide how to weight the gradient that each example produces.

Let $\theta$ be the pre-trained model, and $\mathcal{D}$ be the full training dataset of task $T$. We draw a small subset $\mathcal{D}' \subset \mathcal{D}$ and we fine-tune $\theta$ on $\mathcal{D}'$, collecting each weight update $\Delta\theta_i$ during this process; here, $i$ indexes each batch observed in this fine-tuning run (from 1 to the number of batches in $\mathcal{D}'$). Now, let $k$ represent a linear projection layer in the analysed model, and $\theta^k$ and $b^k$ represent the weights and bias of this layer. We initialise parameters $\alpha_i^k = 1$ which will be used to balance the updates of different $\Delta\theta_i$. We modify the forward pass of every layer to make use of these new $\alpha_i^k$ parameters. Originally, the forward pass of linear layer $k$, when receiving input $x$ is computed as $h^k$:

$$h^k = \theta^k x + b^k \tag{15}$$

We modify this forward pass to:

$$h^k = \theta^k x + b^k + \sum_i \alpha_i^k \left( \Delta\theta_i^k x + \Delta b_i^k \right) \tag{16}$$

At initialisation, i.e., with $\alpha_i^k = 1$, the model's inference is exactly the same as the final model after training on $\mathcal{D}'$. We then freeze the model weights $\theta$, and train parameters $\alpha_i^k$. We train these parameters using the rest of the dataset $\mathcal{D}$ which had not previously been in $\mathcal{D}'$.

Our program $P$, then encodes the subset of the dataset $\mathcal{D}'$ used to collect gradients $\Delta\theta_i$, and the values $\alpha_i^k$ to get to the final version of the model. This combines the data view (encoding a subset of the dataset), and the parametric view (encoding very few parameters) and we hypothesised it could combine their strengths and allow use to push the Pareto frontier.

Figure 11 shows results using this method with SmolLM3 and Olmo3 7B. After extensive experimentation, we concluded that it is not straightforward to push the Pareto frontier with this strategy. It remains unclear whether this indicates the tightness of our Pareto frontier, or just the lack of success of this specific approach. We release these results to encourage future research to target this question.

## H. Behind The Scenes

We provide this section to document the process of arriving at this work, rather than just the final product. This section is written by the first author, to distil their personal learnings during this project.

This project started with the objective of more deeply understanding the SAH. At first, we approached this from an interpretability perspective, by drawing on the Sparse Autoencoders (SAEs) literature (Gao et al., 2025). We attempted to gain insights of how SAE features would change during adaptation. However, this implies training SAEs for different

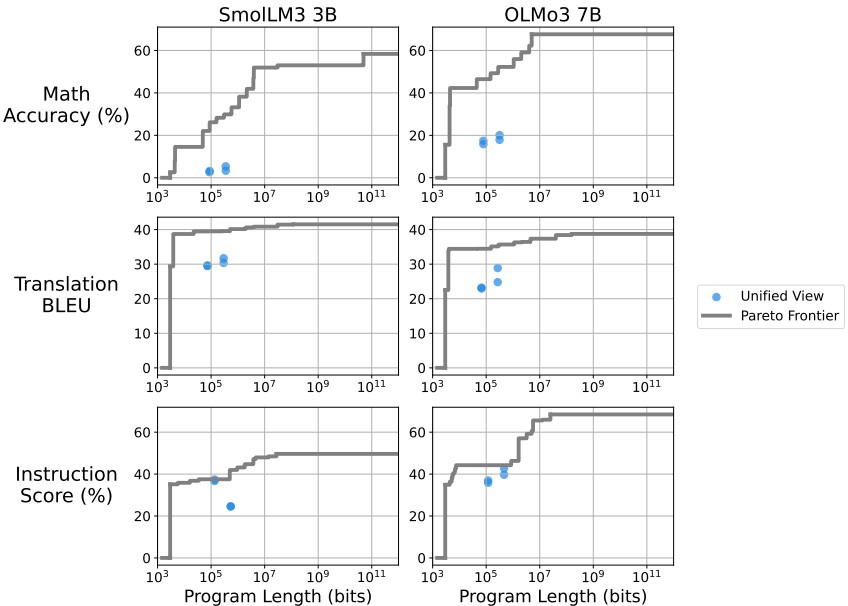

*Figure 11.* Pareto curves in Math (GSM8K), Translation (FLORES) and Instruction (IFEval) compared to our method proposed in Appendix G. We are not able to push the Pareto frontier with our method.

checkpoints of adaptation. After some months of work, we concluded that SAEs were not the right approach to answer our questions. Training SAEs even on the same checkpoint led to vastly different features (Paulo & Belrose, 2026), which completely annulled any attempt at an experimental design to answer our research direction. The first author became increasingly frustrated with the lack of rigor in some of the literature in mechanistic interpretability, and we pushed to more theoretically grounded approaches to understanding the SAH.

Information theory appeared as the right tool to formalise the SAH. However, we were initially drawn to use the perspective of information by Shannon (1948), which the authors were more familiar with. We could not find useful characterisations of the SAH using these definitions. After many months without progress, we landed on a compression perspective, i.e., *How many bits are needed to adapt a model for a task?*. Experiments in this direction led to interesting initial results, which would later become Figure 3. We became interested in formalising such results, and only then related these to Kolmogorov complexity (Kolmogorov, 1965). We realised that it is not this exact notion of complexity, but a lossy version of it, similar to rate-distortion theory (Vereshchagin & Vitányi, 2010). After iterating on definitions, we arrived at our formalisation of task complexity (see Section 3), and the rest of results quickly followed.

Many months of frustration and learning led to the culmination of this work. The main learning from this process is to commit to fundamental questions (such as formalising the SAH), without committing to any specific way of answering them (such as insisting on SAEs or Shannon's notion of information). Once a tool fits the problem (as Kolmogorov complexity did here), the results will follow.

