# OpenReview forum: "Operationalising the Superficial Alignment Hypothesis via Task Complexity"
_ICML.cc/2026/Conference — ICML 2026 regular_

### Official Review · Reviewer_XJBy · 2026-02-27

**Soundness:** 4
**Presentation:** 3
**Significance:** 3
**Originality:** 4
**Overall Recommendation:** 5
**Confidence:** 2

**Summary:**

The Superficial Alignment Hypothesis (SAH), a popular theory in the LLM community suggesting that models learn almost all their capabilities during pre-training, while post-training (alignment) simply exposes these existing capabilities.
The paper addresses the issue of SAH's vague definition. This approach successfully unifies the amount of fine-tuning data, parameter updates, and prompt engineering under a single measurement standard: Task Complexity, defined as the length of the shortest program that achieves a target performance on a task.
This leads to two conclusions:
a. Pre-training, despite achieving relatively low performance, has low task complexity (fundamental).
b. Post-training, while achieving high performance, involves high task complexity (superficial).
The conclusion is aligned with the SAH.

**Compliance With Llm Reviewing Policy:**

Affirmed.

**Key Questions For Authors:**

1. What is the average token length for both the pre-training and post-training models? If the average token length is typically longer in the post-training model, could this introduce a bias related to task complexity?

2. How does the performance or score vary when different maximum response lengths are set in prompts during pre/post-training stages or under different views?

3. How to make sure the fair comparison among different views (data, parametric view, and inference-control)?

**Limitations:**

yes

**Strengths And Weaknesses:**

# Strengths:
1. Formalizing Theory: Transforming Vague Intuition of SAH into Precise Definitions.

2. Unifying Different Points: Unified fine-tuning data, parameter updates, and prompt engineering under a single measurement standard, Task Complexity.

3. The experiments on different tasks and model scales verified the SAH by proposed theory. It maybe a positive support for SAH.

# Weaknesses:
1. It seems the Task Complexity can not be computed accurately in theory.

---

> ### Author Rebuttal · Authors · 2026-03-30
>
> We thank the reviewer for their very positive assessment of our work. We provide the following answers that we hope will help them feel more confident in the value of our paper:
> - **Task complexity is uncomputable, but our experiments provide upper bounds**. We discuss this issue in detail in Section 4. Even considering this, our upper bounds are surprisingly small, which is informative of the degree to which LLM adaptation is superficial in terms of information.
> - **Q1 - What is the average amount of tokens in the responses for both the pre-training and post-training models?**: When evaluating either pre- or post-training models, we set a hyperparameter of 512 tokens generated, and we parse the final answer returned by these models. These response lengths should not have connection to the length of the program that generated these answers, and therefore expect no bias regarding this.
> - **Q2 - How does the performance vary when setting different numbers of tokens for the responses in each setting?**: We set this hyperparameter of 512 tokens in the completion of answers and parse the final one. We do not make any distinction between pre- or post-training models, or the method of adaptation used, when setting the completion length.
> - **Q3 - How to make sure the fair comparison among different views (data, parametric view, and inference-control)?**: We are primarily interested in providing the tightest upper bound for task complexity, which implies that we are interested in representing the programs as concisely as possible in any of the three views. Therefore, if we see a simple way of compressing the program of one of the views (such as using data compression for the data view), it is in the spirit of our research to apply such compression even if it cannot apply to other views. We do make the observation that in our final Pareto curves, we do see the appearance of programs inspired by different views, but this comparison between views is not the main contribution of our work.
>
>
> Again, we thank the reviewer for their positive assessment, and hope these answers help the reviewer feel more confident in the value of our work.

---

> > ### Author Rebuttal · Reviewer_XJBy · 2026-04-01
> >
> > Thank the authors for the responses.
> >
> > Q1. The difference in response may stem from the average token length of training samples during pre-training versus post-training. Typically, pre-training samples are shorter than post-training samples. That may be the reason.
> >
> > Q2. During evaluation, consider setting the maximum tokens as a condition in the prompt, rather than using the direct max_token_length hyperparameter. This approach may help verify whether the post-trained model can achieve better performance with a **similar** token cost.

---

> > > ### Author Response · Authors · 2026-04-03
> > >
> > > We are glad that some of the reviewer’s questions are now addressed. We want to clarify a detail that may be the stem of both of the reviewer’s remaining questions. The response length is not measured as part of the program length, because these responses are not part of the program itself, but as part of the execution. Therefore these response lengths are not modifying the computed task complexities. This is why we expect no bias due to these values, both in terms of the data the models used in pre- vs post-training (Q1), and in terms of the response lengths used in evaluation (Q2). We will clarify this in the revised version of the paper, and appreciate the engagement to help us improve the presentation of our work.

---

### Official Review · Reviewer_yGgV · 2026-03-11

**Soundness:** 2
**Presentation:** 4
**Significance:** 3
**Originality:** 4
**Overall Recommendation:** 3
**Confidence:** 4

**Summary:**

This paper formalizes the Superficial Alignment Hypothesis (SAH) through the lens of Algorithmic Information Theory. The authors introduce the concept of "Task Complexity," defined as the length (in bits) of the shortest adaptation program required to achieve a target performance. Through this framework, the paper unifies data fine-tuning, parametric fine-tuning (LoRA), and inference-time control (Prompting) into a single Pareto frontier of program length versus performance. Experimental results across multiple models (Llama-3, Olmo3) show that pre-trained models often require extremely short programs (less than a few kilobytes) to achieve high performance on downstream tasks, thereby empirically supporting the hypothesis that the alignment process is fundamentally "superficial."

**Compliance With Llm Reviewing Policy:**

Affirmed.

**Final Justification:**

I sincerely appreciate the authors' extensive rebuttal efforts. The paper's originality and significance are strong, offering an intriguing framework to formalize the Superficial Alignment Hypothesis. Furthermore, the newly completed e-SNLI experiments successfully addressed my critical concerns regarding data contamination.However, the rebuttal ultimately reinforced my prior assessment regarding the paper's soundness. The authors' report that full fine-tuning yields a program length of $4.9 \times 10^{10}$ bits (~6GB) perfectly illustrates the structural asymmetry of their core metric. Plotting a Pareto frontier that inherently equates highly compressed text prompts with largely uncompressed neural network weights creates a fundamental measurement bias against parametric methods. Defending this by stating they "are not comparing points" is logically inconsistent with the definition of a Pareto boundary.While the paper is conceptually clear and ambitious, this unresolved metric asymmetry severely distorts the core quantitative claims. Therefore, I maintain my score.

**Key Questions For Authors:**

1. Regarding Measurement Standards: Can the authors provide a set of Pareto curves based on symmetric quantization standards? For instance, either applying an information-theoretic method (e.g., Delta compression) to compress LoRA parameters theoretically, or conversely, using the raw physical bit-width of UTF-8 to measure prompt and data lengths?
2. Regarding Space Complexity: How do the authors reconcile the claim of "superficial adaptation" with the massive dynamic memory overhead (KV Cache) generated when executing long prompts? If runtime space complexity is ignored, is this boundary based strictly on static bit length still theoretically sound?
3. Regarding Data Contamination: Given the data contamination acknowledged in Footnote 6, can the authors provide empirical evidence of "complexity collapse" post-training on entirely new, held-out tasks that were strictly excluded from the post-training mixture?

**Limitations:**

No.
While the authors discuss the theoretical limitation that the uncomputability of Kolmogorov complexity yields only an "upper bound" (Section 8), they fail to critically discuss how their specific "asymmetric compression strategy" and the "ignorance of dynamic inference overhead (KV Cache)" artificially bias this upper bound in favor of prompting methods. It is strongly recommended that the authors add a discussion on how physical system execution overhead limits this theoretical framework.

**Strengths And Weaknesses:**

Strengths:

1. Strong Theoretical Unification (Originality & Significance): Abstracting data, parameters, and prompts into bit-quantized "information payloads" provides a highly novel and elegant mathematical framework to evaluate the long-debated SAH.
2. High Quality of Presentation (Presentation): The paper is exceptionally well-structured and fluidly written. The empirical Pareto frontiers drawn across multiple models and tasks offer strong visual intuition and effectively summarize the macroscopic perspective of model adaptation.

Weaknesses (Primarily concerning Soundness):

1. Inconsistent Quantization Standards (Methodological Flaw): This is the most critical flaw of the paper. When calculating program lengths, the authors apply extreme lossless compression to data and prompts using arithmetic coding based on the model's negative log-likelihood (NLL). In stark contrast, parametric programs (LoRA) are measured using their uncompressed, raw physical bit-width. Comparing "model-prior-compressed text entropy" against "uncompressed physical parameter volume" on the same axis is fundamentally inequitable and artificially inflates the apparent efficiency of the prompting perspective.
2. Ignoring Dynamic Inference Overhead Destroys Theoretical Consistency: The paper compresses prompts into extremely lightweight static text bits while entirely ignoring the spatial complexity explosion when executing them on a physical Turing machine (GPU). In actual inference, a long string of prompts generates massive dynamic state matrices (KV Cache) within the self-attention mechanism, potentially maxing out 80GB of VRAM . Trading extremely high dynamic space complexity (tens of GBs) for extremely low static description length (a few KBs) absolutely cannot be termed "superficial" or "simple" in computational complexity theory.
3. Ignoring Search Cost: The paper explicitly excludes the computational overhead required to discover the optimal adaptation program (e.g., hyperparameter grid search, gradient descent updates). Ignoring the immense logical depth (optimization trajectory complexity) and asserting task superficiality based solely on the final solution's length is highly misleading in its operationalization.
4. Data Contamination in Post-training Evaluation: In Section 7.2, the authors claim that post-training "collapses" the complexity of achieving strong performance. However, Footnote 6 concedes that the GSM8K and IFEval datasets used for testing were actually included in the models' post-training data. Testing on seen distributions fails to prove that the intrinsic complexity of the task was reduced; it merely proves that the program required to "retrieve a memorized answer" is short.

---

> ### Author Rebuttal · Authors · 2026-03-30
>
> We thank the reviewer for their time, and for highlighting the theory we propose. We discuss each point raised as follows:
> - The reviewer raises that applying compression to data-centric approaches but not to parametric approaches creates an unfair comparison of the program lengths obtained. We believe this misrepresents our motivations for this work. We want to estimate the tightest bound of the trade-off between program length and performance obtained, and therefore we try to compress as much as possible any program that we compute.  We are not directly concerned about making judgements between the different approaches, but rather provide general Pareto trade-offs between program length and performance. In the data-centric programs, compressing the data is a technique that we use to achieve as low a program length as possible. However, compressing model weights is far from trivial, and we would be happy to apply such strategy if it were easily applicable.
> - The reviewer argues that ignoring the memory used at inference disregards our theory completely. We believe this claim to be unfounded. We base our theory in algorithmic information theory, in which the main definitions (such as Kolmogorov complexity) are formulated in terms of universal Turing machines. These machines have an infinite tape, and therefore no memory constraints at inference. Therefore, claiming that ignoring these memory constraints “destroys theoretical consistency” is not representative of the algorithmic information theory literature.
> - The reviewer raises that ignoring the search costs to find our programs is misleading of the superficiality of adaptation. We discussed in the paper the fact that Task complexity is in fact uncomputable, making it very difficult to find the absolute minimum program, and therefore we derive several strategies to provide upper bounds for task complexity in practice. The high cost of finding such programs (via doing exhaustive hyperparameter searches and many computed programs) is incurred to provide the best possible bound in task complexity, and which cost is entirely separate from our theoretical definition of task complexity as program length.
> - The reviewer correctly highlights that some mathematical reasoning and instruction following data was used in the post-training of both SmolLM3 and IFEval, which we already discuss in the paper (lines 436 to 439), and which is a confounder for our results in section 7. However, we do have additional results for our translation task (see figure 7), whose dataset was not specifically included in the post-training pipeline of both models, and whose results did not change our conclusions in this section. Additionally, we have now run experiments on e-SNLI (Camburu et. al 2018) as an additional task which has not been explicitly included in the post-training pipelines as additional evidence for our findings. Due to the duration of the hyperparameter exploration to compute the Pareto curves, some training runs are still in progress, but we can already find the same phenomena in the current results. For example, the complexity of getting 70% performance in e-SNLI changes after post-training from 2.96 * 10^5 to 4.26 * 10^3 bits in SmolLM3, and  2.91 * 10^5 to 4.41 * 10^3 bits in Olmo3 7B. We will include the full results of these experiments in the revised version of the paper.
>
>
> We hope these clarifications address the reviewer's concerns regarding the soundness of our contributions.
>
> ---
>
> Camburu, O. M., Rocktäschel, T., Lukasiewicz, T., & Blunsom, P. (2018). e-snli: Natural language inference with natural language explanations. Advances in Neural Information Processing Systems, 31.

---

> > ### Author Rebuttal · Reviewer_yGgV · 2026-04-03
> >
> > I sincerely appreciate the authors' detailed and transparent rebuttal, particularly the theoretical clarifications regarding AIT, which are helpful. However, I continue to have substantial concerns about the empirical execution. The newly provided full fine-tuning program length of $4.9 \times 10^{10}$ bits illustrates my core concern: the metric inherently compares largely uncompressed model weights (~6GB) against highly compressed text, making the "Task Complexity" comparison structurally asymmetric. Furthermore, the acknowledged data contamination  in the core post-training data is a critical issue that compromises the empirical claims. While the preliminary e-SNLI results are a step in the right direction, they are still in progress. Given these foundational empirical issues, I maintain my score.

---

> > > ### Author Response · Authors · 2026-04-03
> > >
> > > We are glad that the concerns about our proposed theory are now resolved.
> > > - Regarding not compressing model weights: we again re-state that our objective is to provide the tightest bound of the Pareto curves of program length and performance. Every program computed (including experiments with compressed text or model weights) is a point in these Pareto curves. We are not focused on comparing these points, but simply used them to compute the tightest estimate of the Pareto curves we can.
> > > - Regarding section 7: we re-state that we already had experiments of this section whose data was not included in post-training of the model (see figure 7). Additionally, all e-SNLI experiments are now finished. The reported results stand.
> > >
> > > We hope these clarifications address the reviewer’s concerns.

---

### Official Review · Reviewer_nmwZ · 2026-03-11

**Soundness:** 2
**Presentation:** 2
**Significance:** 3
**Originality:** 3
**Overall Recommendation:** 5
**Confidence:** 3

**Summary:**

To understand the concept of superficiality in language models, this paper introduces a new metric called Task Complexity, defined as the length of the shortest program that achieves a target level of performance on a given task. The authors formalise the superficial alignment hypothesis (SAH) as the claim that access to a pre-trained model can drastically reduce the complexity of achieving high performance on many tasks that would otherwise be highly complex. Using this framework, the paper unifies several existing perspectives supporting SAH, including data-based, parametric, and inference views, by interpreting them as different strategies for constructing short programs that achieve strong task performance.

**Compliance With Llm Reviewing Policy:**

Affirmed.

**Final Justification:**

The authors' rebuttal addresses my concerns.

**Key Questions For Authors:**

Regarding parametric methods. What if the full-fine tuning is used? Full fine-tuning would likely result in a longer program length (due to storing updated weights), while potentially achieving higher performance. Examining this case could provide additional insight into the trade-off between program length and task performance.

**Limitations:**

See above.

**Strengths And Weaknesses:**

Strengths
- The core idea is interesting and conceptually appealing. The notion of measuring task difficulty via program length provides a potentially useful theoretical lens for analysing language model capabilities.
- The paper provides comprehensive definitions and attempts to formalise discussions around superficial alignment previously. Experiments and analysis are conducted across three different task domains, including mathematical reasoning, machine translation, and instruction following, and include models of varying sizes.



Weaknesses
- The key concept of the paper is the length of the program, yet the computation method is mainly presented in the appendix. Providing a detailed explanation of the calculation method in the main text would improve this paper’s clarity.
- The current organisation of the paper is somewhat difficult to follow. For example, (i) Section 2, which introduces three perspectives on superficial adaptation, appears somewhat disconnected from the subsequent development in Section 3; (ii) At the end of Section 3 (which primarily introduces formal definitions), there is a subsection discussing connections to prior work, which feels structurally misplaced; and (iii) Section 4 again introduces the three views when discussing the estimation of task complexity, leading to some redundancy. A clearer separation between conceptual framing, formal definitions, and empirical estimation would likely improve readability.
- When measuring the length of the program, the implementation does not account for the code contained in external libraries. This could introduce variability depending on the programming environment and assumptions about what constitutes the program.
- More importantly, the program length consists of three components: data, model weights, and code file. The relative contribution of these components could affect the final complexity estimate and the resulting findings. However, the paper does not provide a quantitative analysis of these proportions.

---

> ### Author Rebuttal · Authors · 2026-03-30
>
> We thank the reviewer for the positive assessment of our work, and valuable feedback which we have used to further clarify and strengthen our contribution. We have made the following changes
> - **We extended the explanation of how to compute the size of a program in the main section of the paper.** Due to space constraints, we left many details in appendix A, also raised by reviewer mm31. Based on both reviewer’s comments, we dedicate a considerable portion of the extra page allowed for the camera ready to expand on this in the main section of the paper, while also providing examples of programs and their sizes.
> - **We have improved the connections between sections to improve the readability of the paper.** In particular, in Section 2 we have explicitly mentioned that each view of superficiality can be interpreted as a short program to adapt a model, to give a more natural transition to the definitions based in algorithmic information theory in the following section. In Section 4, we refer back to the argument of each view as a short program to reduce redundancy with respect to Section 2. We hope these changes address the reviewer’s concerns with respect to readability of the paper and we are open to discuss additional suggestions  from the reviewer for how to further improve the readability of our work.
> - **We have significantly improved our foundation for considering Python libraries as part of the machine (and therefore not measured in the program length).** We acknowledge the reviewer’s concern that the decision to consider Python libraries as part of the machine (and therefore not measured in terms of program length) is not extensively justified, and we therefore expand our foundation for making such decision:
>   - We have added a proof that the invariance theorem of Kolmogorov complexity extends to Task complexity. This means that any two universal Turing machines have comparable measured task complexities (bounded by a constant difference that does not depend on the task itself, but rather exclusively on the machines). This implies that whether we allow Python libraries to be part of the machine or not, the difference in Task complexity values is bounded by a constant, making this particular choice of universal Turing machine comparable to the Turing machine without Python libraries.
>   - Even if any universal Turing machine seems like an appropriate choice due to the invariance theorem, we aim for an operationalisation that is as close as possible to the actual programs and machines that run language models. In practice, any real program designed to adapt and run language models will have access and use Python libraries, and do not need to re-write all the utilities defined by these. We argue that considering these Python libraries as part of the machine is much closer to the real machines as they will have built in libraries (such as math) and very likely to already have common libraries (such as numpy or the huggingface utilities).
>
>
> - **We clarify the relative contribution of data, model weights, and code file to our measured program lengths.** We now compute the relative contribution of the code files to the computed program lengths. When computing the relative contribution of the code file in the final length, in the inference-control strategies, ICL and URIAL, we find substantial contributions, with median values of 73.2% and 86.5% respectively. This is due to these strategies having the shortest program lengths, and therefore the script size has a larger relative contribution. In comparison, the Subset Training, LoRA and Bayesian-LoRA strategies have a median contribution of 0.4%, 0.0%, and 0.2% respectively, which indicates that the overall program size becomes dominated by the size of data (in the subset training strategy) or size of parameters (in the case of LoRA based approaches). We add a plot of these distributions to the Appendix.
> - **We add additional experiments using full fine-tuning to further explore the trade-off between program length and performance.** We run these experiments on both SmolLM3 and Olmo3 7B for all of our tasks. We find in practice that the performance of LoRA is generally very similar to that of full fine-tuning, placing many of these experiments outside the Pareto frontier. However, we find cases that do push the frontier further, such as SmolLM3 for GSM8K, where we achieve 58.4% accuracy at a program length of $4.9 * 10^{10}$ bits. We will add these new experiments to the camera ready version.
>
>
> Again, we appreciate the positive review, and valuable feedback. We hope that these new changes allow the reviewer to feel even more confident in the value of our contributions.

---

> > ### Author Rebuttal · Reviewer_nmwZ · 2026-04-03
> >
> > Thanks for your rebuttal. I decided to raise my scores.

---

### Official Review · Reviewer_mm31 · 2026-03-12

**Soundness:** 4
**Presentation:** 3
**Significance:** 3
**Originality:** 3
**Overall Recommendation:** 5
**Confidence:** 3

**Summary:**

The paper introduces an information-theoretic framework to test SAH, defining task complexity as the length of the shortest program needed to achieve a given performance level. It shows that pre-training contains most of the knowledge, while post-training mainly reduces the complexity needed to access it.

**Compliance With Llm Reviewing Policy:**

Affirmed.

**Final Justification:**

The authors addressed all my concerns.

**Key Questions For Authors:**

1. What is the program P used in the experiments? Is P a Python script, and if so, is it generated by the language model itself or constructed separately? It would be very helpful to see a concrete, end-to-end worked example: what the input data looks like, what the program contains at different stages, what the output is, and how the resulting complexity (program length in bits) is actually measured.

2. How would the code complexity change without allowing the use of Python libraries? I feel that using Python libraries can bias the results.

3. Could hybrid or search-based approaches for generating short adaptation programs push the Pareto frontier further?

4. Have the authors considered evaluating tasks with exact, formally verifiable metrics, like ListOps or just simple modulo arithmetic operations, where the link between program structure and performance can be precisely measured? Or can you imagine any small toy example where we have an exact solution?

5. A single task can have many different solutions (e.g., modular addition via a lookup table or an algorithm). If two programs have the same length, are they treated as equally complex, even if the strategies differ?

**Limitations:**

yes

**Strengths And Weaknesses:**

- **Soundness and Presentation:** The paper sounds technical, and it is well written. The background and the motivation are well structured. I really enjoyed reading the paper. The methodology paper could be strengthened to clarify how exactly the programs (P) are created; it is not clear to me. The results, however, are well presented, and the Pareto curves effectively communicate the central empirical message.

- **Significance and Originality:**  I think the paper is significant and addresses an important gap in the literature. Formalizing the SAH through the lens of task complexity is a non-trivial contribution.  The unification of the data, parametric, and inference-control views under a single quantitative framework is conceptually clean and offers a new way of thinking about what adaptation actually costs in information-theoretic terms. The finding that post-training collapses task complexity by orders of magnitude is both surprising and interpretively rich. This is a research direction with clear potential to influence how the community reasons about pre-training, alignment, and efficient adaptation going forward.

---

> ### Author Rebuttal · Authors · 2026-03-30
>
> We thank the reviewer for their valuable feedback, and we are encouraged by their positive evaluation of the potential impact of our work. Based on the reviewer’s comments, we have added modifications that we hope will help the reviewer feel even more confident in the value of our work. These are:
> - **Q1 - What is a program P used in the experiments?**: Any program P used in our experiments consists of a python script (not created by the model), and any data or parameters used in the script. We include the details of how to measure the length of such a program in Appendix A due to space constraints. We will use the extra page of the camera ready to provide a more detailed explanation in the main body of the paper. We will additionally include examples of our python scripts, and the measured sizes of parameters and data used, to provide a much clearer picture to the reader.
> - **Q2 - How would the task complexity values change if we disallowed using Python libraries?**: Disallowing the use of Python libraries would result in longer measured task complexities, as the programs would need to re-implement the used functionality. This implies that our complexity curves get shifted, but its overall shape should remain unaffected. We expand our discussion on why we chose this operationalisation:
>    - We have added a proof that the invariance theorem of Kolmogorov complexity extends to Task complexity. This means that any two universal Turing machines have comparable measured task complexities (bounded by a constant difference that does not depend on the task itself, but rather exclusively on the machines). This implies that whether we allow Python libraries to be part of the machine or not, the difference in Task complexity values is bounded by a constant, making this particular choice of universal Turing machine comparable to the Turing machine without Python libraries.
>   - Even if any universal Turing machine seems like an appropriate choice due to the invariance theorem, we aim for an operationalisation that is as close as possible to the actual programs and machines that run language models. In practice, any real program designed to adapt and run language models will have access and use Python libraries, and do not need to re-write all the utilities defined by these. We argue that considering these Python libraries as part of the machine is much closer to the real machines as they will have built in libraries (such as math) and very likely to already have common libraries (such as numpy or the huggingface utilities).
> - **Q3 - Could hybrid or search-based approaches for generating even shorter programs?**: We appreciate that the reviewer shares this intuition, and in fact in Appendix G, we experimented with a hybrid approach between the parametric and the data view. We hoped that combining both views would allow us to push the Pareto frontier but we got a negative result. Nevertheless, we hope that our method provides a starting point for others and hence we included it in the Appendix.
> - **Q4 - Have the authors considered evaluating tasks where the link between task complexity and the construction of the task itself can be provably measured?**: We considered including (toy) tasks (such as ListOps) where we could theoretically derive a lower bound for task complexity, and believe this to be possible and an interesting research direction. Ultimately, we decided to  focus on tasks already used in the superficial alignment literature, such as instruction following, which was present in the original paper proposing the hypothesis, and GSM8K, which was used in the critiques to the SAH which we revisited in section 6.3.
> - **Q5 - If two programs have the same length, are they considered equally complex?**: Our notion of complexity is based on the original foundations of Algorithmic Information Theory, in which only the program length is considered. So, yes, any two programs with the same length will be considered equally complex.
>
>
> Again, we thank the reviewer for their positive assessment, and hope these revisions address the reviewer's concerns and further strengthen the paper.

---

> > ### Author Rebuttal · Reviewer_mm31 · 2026-04-02
> >
> > I thank the authors for their time and effort in addressing my concerns. I will increase my score.

---

### Decision · Program_Chairs · 2026-04-30

**Decision:**

Accept (regular)

**Comment:**

The paper provides a highly original, conceptually elegant formalization of the Superficial Alignment Hypothesis (SAH) using Algorithmic Information Theory. By redefining adaptation cost as "Task Complexity" (the shortest program length required to achieve target performance), it successfully unifies data, parametric, and prompt-based adaptation methods into a single Pareto frontier. The observation that post-training collapses the complexity of reaching strong performance by orders of magnitude is both surprising and interpretively rich for the community.

During the discussion, a highly rigorous theoretical debate took place. Reviewer yGgV raised a critical soundness concern regarding the asymmetrical measurement of complexity—specifically, applying heavy information-theoretic compression to text/prompts while measuring parametric updates (LoRA) by their uncompressed raw physical bit-width, thereby inherently biasing the Pareto frontier against parametric methods. The authors defended this by grounding their methodology in universal Turing machine assumptions and the pursuit of the tightest possible bounds. While the committee acknowledges that this measurement asymmetry remains a structural limitation of operationalizing algorithmic information theory for continuous neural weights, it does not invalidate the paper's conceptual contribution. Furthermore, the authors did an excellent job addressing data contamination concerns by running new experiments on the e-SNLI dataset during the rebuttal, solidifying their empirical claims.

Overall, this is a technically solid, thought-provoking paper that introduces a novel theoretical lens to understanding LLM alignment. It will spark valuable discussion at the conference.